# Frame Context Packing and Drift Prevention in Next-Frame-Prediction Video Diffusion Models

**Lvmin Zhang**[1]   **Shengqu Cai**[1]   **Muyang Li**[2]   **Gordon Wetzstein**[1]   **Maneesh Agrawala**[1]

[1]Stanford University   [2]MIT

{lvmin,shengqu,gordon.wetzstein,maneesh}@cs.stanford.edu, muyangli@mit.edu

## Abstract

We present a neural network structure, FramePack, to train next-frame (or next-frame-section) prediction models for video generation. FramePack compresses input frame contexts with frame-wise importance so that more frames can be encoded within a fixed context length, with more important frames having longer contexts. The frame importance can be measured using time proximity, feature similarity, or hybrid metrics. The packing method allows for inference with thousands of frames and training with relatively large batch sizes. We also present drift prevention methods to address observation bias (error accumulation), including early-established endpoints, adjusted sampling orders, and discrete history representation. Ablation studies validate the effectiveness of the anti-drifting methods in both single-directional video streaming and bi-directional video generation. Finally, we show that existing video diffusion models can be finetuned with FramePack, and analyze the differences between different packing schedules.

## 1   Introduction

Forgetting and drifting are the two most critical problems in next-frame-prediction models for video generation. "Forgetting" refers to the fading of memory as the model struggles to remember earlier content and maintain consistent temporal dependencies. "Drifting" refers to the degradation of visual quality due to error accumulation over time (also called exposure/observation bias).

A fundamental dilemma emerges when attempting to simultaneously address both forgetting and drifting: any method that mitigates forgetting by enhancing memory may also increase error accumulation/propagation, thereby exacerbating drifting; any method that reduces drifting by interrupting error propagation needs to weaken temporal dependencies (*e.g.*, masking or re-noising the history), thus worsening the forgetting. This is a fundamental trade-off that hinders the scalability of next-frame prediction models.

A naive solution to forgetting is to encode more frames. But this approach quickly becomes computationally intractable due to the quadratic attention complexity of transformers (even with optimizations like Flash Attention [10], *etc*.). Moreover, video frames contain significant temporal redundancy, making naive full-context approaches very inefficient. The substantial duplication of visual features across consecutive frames suggests the potential to design effective compression systems to facilitate memorization.

Drifting is influenced by memorizing mechanisms in several ways. The source of drifting lies in the initial errors that occur in individual frames, while the effect is the propagation and accumulation of these errors across subsequent frames, and the eventual drifting out of the train distribution. A stronger memorization mechanism, on one hand, can lead to better temporal consistency and reduce the occurrence of initial errors. On the other hand, it also memorizes more errors and thus accelerates error propagation when errors do occur. This paradoxical relationship between memory mechanisms

39th Conference on Neural Information Processing Systems (NeurIPS 2025).

and drifting necessitates carefully designed training/sampling methods to facilitate error correction or interrupt error propagation.

In this paper, we propose FramePack as an anti-forgetting memory structure along with anti-drifting sampling and training methods. The FramePack structure addresses the forgetting problem by compressing input frames based on their relative importance, ensuring that the total transformer context length converges to a fixed upper bound. We consider both time-proximity-based and feature-similarity-based importance measures. Afterwards, we propose anti-drifting sampling methods that break the causal prediction chain and incorporate bi-directional contexts by planning single or multiple endpoint frames. We also present an anti-drifting training method to convert frame history into discrete tokens so as to reduce the history disparity between training and inference. We show that these methods effectively reduce the occurrence of errors and prevent their propagation.

We demonstrate that existing pretrained video diffusion models (*e.g.*, HunyuanVideo [28], Wan [48], *etc.*) can be finetuned with FramePack. Our experiments reveal several findings: because next-frame prediction generates smaller tensor sizes per step compared to full-video generation, it enables more balanced diffusion schedulers with less extreme flow shift timesteps. We also show that the efficient implementations of FramePack can process thousands of frames with 13B models even on laptops (*e.g.*, 6GB or 8GB GPU memory).

## 2 Related Work

### 2.1 Anti-forgetting and Anti-drifting

The trade-off between forgetting and drifting is also evidenced by previous discussions. CausVid [65] shows that when the video generator is causal, the quality degradation appears at the end of the video and the high-quality part may be subject to an upper bound length. DiffusionForcing [6] discussed that the cause of this drift may be related to error accumulation in models' observation disparity between training and inference. Wang *et al.* [51] discussed that a model with stronger memory may suffer more from drifting and error accumulation.

**Noise scheduling and augmentation in history frames** modify noise levels at specific timesteps, video times, or image frequencies to mitigate drifting. These methods generally reduce the dependency on past frames. DiffusionForcing [6] and RollingDiffusion [39] are typical examples. Our ablation studies investigate the influence of adding noise to history frames.

**Classifier-Free Guidance (CFG) over history frames** applies different masks or noise levels to opposite sides of guidance to amplify the forgetting-drifting trade-off. HistoryGuidance [41] demonstrates this approach. Our ablation studies include guidance-based noise scheduling.

**Anchor frames** can be used as planning elements for video generation. StreamingT2V [20] and ART-V [54] use reference images as anchors. Video planning approaches [32, 73, 22, 60, 3, 61] use image or video anchors for content planning.

**Compressing latent space** can improve the efficiency of video diffusion models. FlexTok [2] adjusts token context length to achieve different levels of visual content compression. LTXVideo [17] shows that a highly compressed latent space can be used for diffusing videos efficiently. PyramidFlow [25] diffuses video latents in a pyramid and re-noises downsampled latents in that pyramid to reduce computation costs. FAR [16] proposes a multi-level causal attention structure to establish long-short-term causal context pacifying and KV caches. HiTVideo [76] uses hierarchical tokenizers to enhance the video generation with autoregressive language models.

**Memory in world models** often involves different modeling of long-term memory. Typical examples are 3D geometry like mesh and proxy [38, 50, 67, 66]. Training diffusion models with domain data can also bake the memory into the model, with full model training [1, 46] or low-rank methods [21]. Retrieval-based memory mechanisms like WorldMem [58] are efficient when the task prioritizes reconstructing history contents.

### 2.2 Long Video Generation

Extending video generation beyond short clips remains an open problem. LVDM [19] generates long videos using latent diffusion, while Phenaki [47] creates variable-length videos from sequences

of text prompts. Gen-L-Video [49] applies temporal co-denoising for multi-text conditioned videos, and FreeNoise [37] extends pretrained models without additional training via noise rescheduling. NUWA-XL [62] implements a Diffusion-over-Diffusion architecture with coarse-to-fine processing, while Video-Infinity [44] overcomes computational constraints through distributed generation. StreamingT2V [20] produces consistent, dynamic, and extendable videos without hard cuts, and CausVid [65] transforms bidirectional models into fast autoregressive models through distillation. Recent advances include GPT-like architecture (ViD-GPT [15]), multi-event generation (MEVG [36]), attention control for multi-prompt generation (DiTCtrl [5]), precise temporal control (MinT [55]), history-based guidance (HistoryGuidance [41]), unified next-token and full-sequence diffusion (DiffusionForcing [6]), SpectralBlend temporal attention (FreeLong [33]), video autoregressive modeling (FAR [16]), and test-time training (TTT [9]). Harvey *et al.* [18] proposes a flexible approach for modeling long contexts with dilatation (Hierarchy-2) and propagation. Generating longer videos often requires efficient architectures, *e.g.*, linear attention [4, 59, 52, 8, 68, 26], sparse attention [56, 71, 72, 57], low-bit computation [30, 74, 29], low-bit attention [70, 69], hidden state caching [35, 31], distillation [42, 34, 63, 64], *etc*.

## 3 Packing Frame Context

We consider a video generation model that predicts next frames repeatedly to form a video. For simplicity, we consider next-frame-section prediction models using Diffusion Transformers (DiTs) that generate a section $X$ of $S$ unknown frames so that $X \in \mathbb{R}^{S \times h \times w \times c}$, conditioned on a section $F$ of $T$ input frames so that $F \in \mathbb{R}^{T \times h \times w \times c}$. All definitions of frames and pixels refer to latent representations, as most modern models operate in latent space.

For next-frame (or next-frame-section) prediction, $S$ is typically 1 (or a small number). We focus on the challenging case where $T \gg S$. With per-frame context length $L_f$ (typically $L_f \approx 1560$ for each 480p frame in Hunyuan/Wan/Flux), the vanilla DiT yields total context length $L = L_f(T + S)$. This causes a context length explosion when $T$ is large. We observe that the input frames have different importance when predicting the next frame, and we can prioritize them according to their importance.

### 3.1 Time Proximity Based Packing

We first consider a baseline case where the temporal proximity reflects frame importance (Fig. 1-(a)). More advanced cases involving similarity-based importance are covered in §3.2. With frames temporally closer to the prediction target being more relevant, we enumerate all frames with $F_0$ being the most important (*e.g.*, the most recent) and $F_{T-1}$ being the least (*e.g.*, the oldest). We define a length function $\phi(F_i)$ that determines each frame's context length after VAE encoding and transformer patchifying by applying progressive compression

$$\phi(F_i) = \frac{L_f}{\lambda^i} \, , \tag{1}$$

where $\lambda > 1$ is a compression parameter. The frame-wise compression is achieved by manipulating the transformer's patchify kernel size in the input layer (*e.g.*, $\lambda = 2, i = 5$ means a kernel size where the product of all dims equals $2^5 = 32$ like the 3D kernel $2 \times 4 \times 4$, or $8 \times 2 \times 2$, *etc.*). The total context length then follows a geometric progression

$$L = S \cdot L_f + L_f \cdot \sum_{i=0}^{T-1} \frac{1}{\lambda^i} = S \cdot L_f + L_f \cdot \frac{1 - 1/\lambda^T}{1 - 1/\lambda} \, , \tag{2}$$

and when $T \to \infty$, the total context length converges to $\lim_{T \to \infty} L = (S + \frac{\lambda}{\lambda-1})$. This bounded context length makes FramePack's compression bottleneck invariant to the input frame number $T$.

Since most hardware supports efficient matrix processing by powers of 2, we mainly discuss the case of $\lambda = 2$ in this paper. Note that we can represent arbitrary compression rates by duplicating (or dropping) several specific terms in the power-of-2 sequence: considering the accumulation $\sum_{i=0}^{+\infty} \frac{1}{2^i} = \frac{2}{2-1} = 2$, if we want to loosen it a bit, for example to 2.625, we can duplicate the terms $\frac{1}{2}$ and $\frac{1}{8}$ so that $\frac{1}{1} + \frac{1}{2} + (\frac{1}{2}) + \frac{1}{4} + \frac{1}{8} + (\frac{1}{8}) + \frac{1}{16} + ... = \frac{1}{2} + \frac{1}{8} + \sum_{i=0}^{+\infty} \frac{1}{2^i} = 2.625$. Following this, one can cover arbitrary rates by converting the rate value to binary bits and then translating every bit.

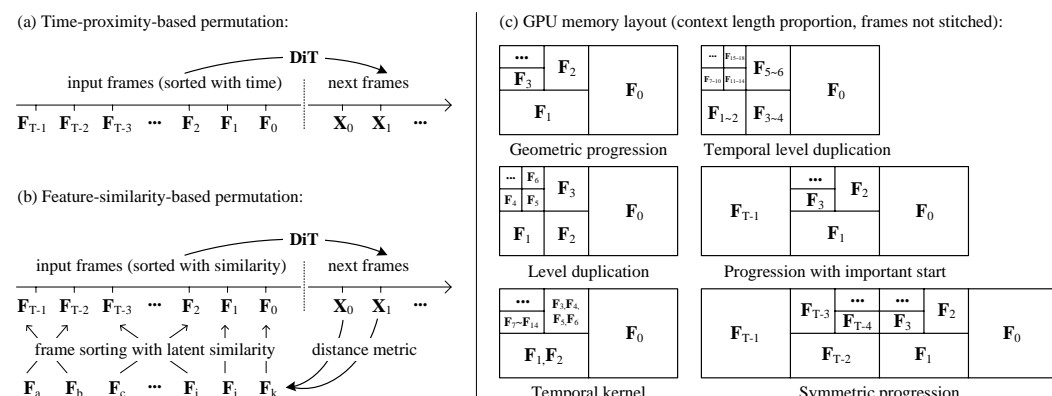

Figure 1: **FramePack variants.** We present frame packing methods using time proximity or feature similarity. We discuss several typical kernel structures. This list does not necessarily cover all popular variants, and more structures can be developed in a similar way.

**Packing schedules** The patchifying operations in most DiTs are 3D, and we denote the 3D kernel as $(p_f, p_h, p_w)$ representing the steps in frame number, height, and width. A same compression rate can be achieved by multiple possible kernel sizes, *e.g.*, the compression rate of $64$ can be achieved by $(1, 8, 8)$, $(4, 4, 4)$, $(16, 2, 2)$, $(64, 1, 1)$, *etc*. Compression levels can be duplicated and combined with higher compression rates. We discuss more packing ways in Fig. 1-(c): *duplication* allows for same kernel sizes in frame width and height, making the compression more compact; *temporal kernel* can compress contiguous frames into a single tensor; *symmetric progression* treats both beginning and ending frames as equally important.

**Independent patchifying parameters** Empirical evidence shows that using independent parameters for the different input projections at multiple compression rates facilitates stabilized learning. We assign the most commonly used input compression kernels as independent neural network layers: $(2, 4, 4)$, $(4, 8, 8)$, and $(8, 16, 16)$. For higher compressions (*e.g.*, at $(16, 32, 32)$), we first downsample (*e.g.*, with $2 \times 2 \times 2$) and then use the largest kernel $(8, 16, 16)$. We initialize their separated weights by interpolating from the pretrained patchifying projection (*e.g.*, the $(2, 4, 4)$ projection of HunyuanVideo/Wan).

**Tail options** While in theory FramePack can process videos of arbitrary length with a fixed, invariant context length, frames may fall below a minimum unit size (*e.g.*, a single latent pixel) when the input length becomes extremely large. We discuss 3 options to process the tail frames: (1) simply delete the tail; (2) allow each tail frame to increase the context length by a single latent pixel; (3) apply global average pooling to all tail frames and process them with the last kernel. In our tests, the visual differences between these options are relatively negligible.

**RoPE alignment** When encoding inputs with different compression kernels, the different context lengths require RoPE (Rotary Position Embedding) [43] alignment. RoPE generates complex numbers with real and imaginary parts for each token position across all channels, which we refer to as "phase". We directly downsample (using average pooling) such phases to match the compression kernels.

### 3.2 Feature Similarity Based Packing and Hybrid Approach

The aforementioned frame ordering $\boldsymbol{F}_{0\ldots T-1}$ can be seen as a result of sorting all history frames using their time positions. We note that such sorting can also use other metrics like feature similarity (Fig. 1-(b)). For instance, consider a typical cosine similarity

$$\text{sim}_{\text{cos}}(\boldsymbol{F}_i, \hat{\boldsymbol{X}}) = \sum_p \frac{(\boldsymbol{F}_i)_p \cdot \hat{\boldsymbol{X}}_p^\top}{\|(\boldsymbol{F}_i)_p\| \|\hat{\boldsymbol{X}}_p\|} \,, \tag{3}$$

where the sum is taken over pixels $p$. This measures the cosine similarity between each history frame and the estimated next frame section. Sorting the history frames using $\text{sim}_{\text{cos}}(\cdot)$ will produce a

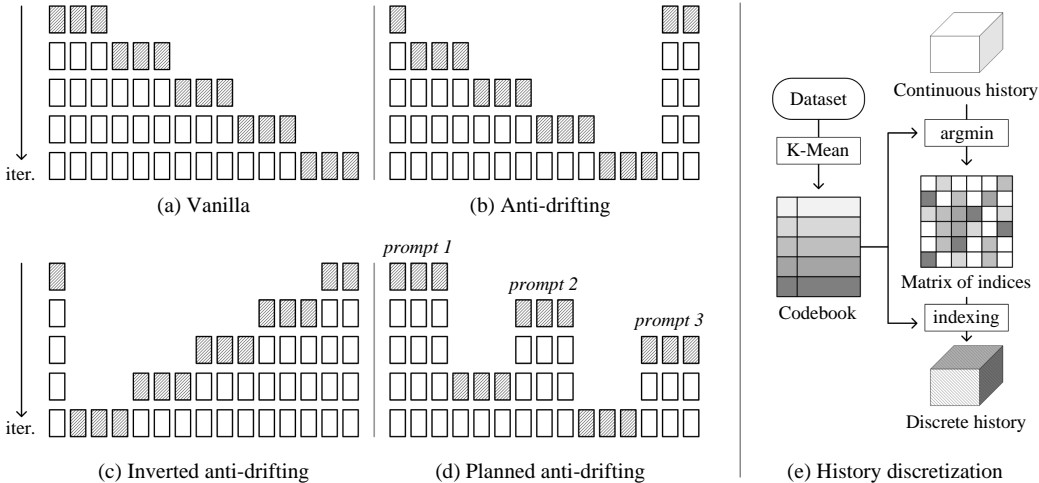

(a) Vanilla       (b) Anti-drifting

(c) Inverted anti-drifting    (d) Planned anti-drifting    (e) History discretization

Figure 2: **Anti-drifting sampling and training methods.** We present sampling approaches to generate frames in different temporal orders. The shadowed squares are the generated frames in each iteration, whereas the white squares are the iteration inputs. We also discuss the method to convert the frame history into a discrete representation.

permutation $\boldsymbol{F}_{0\ldots T-1}$ with $\boldsymbol{F}_0$ being the most similar and $\boldsymbol{F}_{T-1}$ being the least. This permutation can directly replace the aforementioned time proximity. Since similarity-based permutation may change abruptly when processing contiguous frames, we also consider a smooth time proximity modeling

$$\mathrm{sim}_{time}(\boldsymbol{F}_i, \hat{\boldsymbol{X}}) = e^{-(\mathrm{time}(\boldsymbol{F}_i) - \mathrm{time}(\hat{\boldsymbol{X}}))^2} , \qquad (4)$$

where $\mathrm{time}(\cdot)$ gets the frames' starting time measured in seconds. Consider the weighting

$$\mathrm{sim}_{\mathrm{hybrid}}(\boldsymbol{F}_i, \hat{\boldsymbol{X}}) = \mathrm{sim}_{\mathrm{cos}}(\boldsymbol{F}_i, \hat{\boldsymbol{X}}) + \lambda_{time}\, \mathrm{sim}_{time}(\boldsymbol{F}_i, \hat{\boldsymbol{X}}) , \qquad (5)$$

where $\lambda_{time}$ is a weighting parameter. Sorting the history frames using $\mathrm{sim}_{\mathrm{hybrid}}(\cdot)$ will produce a permutation that transits relatively smoothly as the generating window moves forward in time. This hybrid approach is suitable for world model datasets (mainly video games) that require returning to previously visited views of scenes or events. Similar sorting can also be applied to facial identity metrics to facilitate movie generation applications that emphasize consistent human actors. This method will be evaluated in ablation experiments in the supplementary materials.

## 4 Drift Prevention

Drifting is a common problem in next-frame prediction models where visual quality degrades as video length increases. We discuss anti-drifting approaches by adjusting the sampling processes and history representations as shown in Fig. 2.

### 4.1 Planned Endpoints and Adjusted Sampling Order

One possible explanation of drifting is that the modeling of the chained conditional probability $\mathbb{P}(\boldsymbol{X}_t|\boldsymbol{X}_{t-1})$ fails to approximate $\mathbb{P}(\boldsymbol{X}_t)$ due to imperfect estimations. This perspective indicates that a strict causal system is more vulnerable to drifting than bi-directional models that directly approximate $\mathbb{P}(\boldsymbol{X}_t|\boldsymbol{X}_{t_1}, \boldsymbol{X}_{t_2})$ where $t_1 < t < t_2$.

**Endpoint planning** The vanilla sampling method shown in Fig. 2-(a) can be modified into Fig. 2-(b), where the first iteration simultaneously generates both beginning and ending sections, while subsequent iterations fill the gaps between these anchors. This simple method can get rid of drifting in specific cases when the video motions are in a relatively small range, or when the motion patterns are repeated or periodic (*e.g.*, dancing, talking, spinning, *etc.*), or when the motion content follows some texture patterns (*e.g.*, fire flame, water flow, *etc.*).

**Inverted sampling (image-to-video)**  A variant by inverting the sampling order in Fig. 2-(b) into Fig. 2-(c) is effective for image-to-video generation. In image-to-video, the first frame is a ground-truth user input, whereas the last frame is a generated endpoint that is not guaranteed to perfectly preserve the quality of the user input. All generations in Fig. 2-(c) keep the direction to approximate the high-quality user input, leading to iteratively refined generations.

**Multiple endpoints**  The endpoint planning can be repeated with different prompts before filling in the gaps, resulting in a planned sequence of generation (Fig. 2-(d)). Note that though the drifting might still happen in the endpoint-wise, in certain cases, when the prompted sections are distant enough, the error accumulation becomes almost negligible. This method is more flexible than single endpoint planning and supports more dynamic motions and more complicated storytelling.

**RoPE with random access**  These sampling methods require modifications to RoPE to support non-consecutive phases (time indices of frames). This is achieved by skipping the blank phases (indices) in the time dimension.

## 4.2  History Discretization

Another potential cause of drifting is the difference between training and inference distributions over the history frames. This indicates that drifting can be mitigated if the history representation is not sensitive enough to distinguish between the training and inference frames. Discrete integer tokens are well-suited to reduce the mode gap between training and inference distributions. This perspective is supported by empirical evidence that discrete autoregressive systems (*e.g.*, LLMs) often demonstrate less obvious drifting than continuous autoregressive systems for visual content diffusion.

We discretize the history as in Fig. 2-(e). Consider a dataset $\Phi \in \mathbb{R}^{(B \times T \times H \times W) \times C}$ of precomputed latent videos, a $K$-Mean over all latent pixels will yield a codebook $\Omega \in \mathbb{R}^{K \times C}$ where $K \in \mathbb{Z}^+$ is an adjustable number. Any latent frame $\boldsymbol{F} \in \mathbb{R}^{T \times H \times W \times C}$ can be quantized by $Q(\cdot)$ with

$$Q(\boldsymbol{F})_p = \arg \min_k ||\boldsymbol{F}_p - \Omega_k||_2 \,, \tag{6}$$

where $p$ is pixel position, and $Q(\boldsymbol{F}) \in [0, K-1]^{T \times H \times W}$ is a matrix of indices over the codebook $\Omega$. The matrix of indices can be converted back to latent videos by $\Omega_{Q(\boldsymbol{F})} \in \mathbb{R}^{T \times H \times W \times C}$. We replace all history frames $\boldsymbol{F}$ with $\Omega_{Q(\boldsymbol{F})}$ during training.

Intuitively, when $K = 1$, the history becomes meaningless as one single color, and the drifting is eliminated at the cost of giving up memory (errors do not accumulate but sections become unrelated); when $K \to \infty$, the effect is equivalent to no discretization, and the drifting remains. We show with ablation study that a suitable $K$ can minimize the error propagation while simultaneously producing plausible consistency between sections.

# 5  Experiments

## 5.1  Ablative Naming

To simplify the presentation of the experiments, we use a common naming convention for all ablative structures. A FramePack name is represented as a string such as `td_f16k4f4k2f1k1_g9_x_f1k1`. We explain the meaning of this notation:

*Kernel*: A kernel name is like `k1h2w2`. The `k` stands for "kernel", and `k1h2w2` indicates a patchify kernel with shape $(1, 2, 2)$, where the temporal size is 1, the height is 2, and the width is 2.

*Kernel (simplified)*: For simplicity, since kernels that are multiples of $(1, 2, 2)$ are commonly used, we use abbreviated notation such as `k1` that only denotes the temporal dimension. Specifically, `k1` represents `k1h2w2` (the kernel $(1, 2, 2)$), `k2` represents `k2h4w4` (the kernel $(2, 4, 4)$), *etc*.

*Encoding frames*: The notation `f16k4` indicates that 16 frames are encoded by the kernel `k4` (the simplified `k4h8w8`) with kernel size $(4, 8, 8)$.

*Packing*: The notation `f16k4f2k2f1k1` shows a way to encode 19 contiguous frames: the first 16 frames are encoded by the kernel `k4` (the kernel $(4, 8, 8)$), the next 2 frames are encoded by the kernel `k2` (the kernel $(2, 4, 4)$), and the last 1 frame is encoded by the kernel `k1` (the kernel $(1, 2, 2)$).

*Tail*: We append the notation with `td`, `ta`, or `tc` to indicate the tail frames before or after packing, such as `td_f16k4f4k2f1k1`. The three options are as discussed in Section 3.1. Herein, the "delete" option `td` deletes the tail. The "append" option `ta` compresses each tail frame by performing a 3D pooling of $(1, 32, 32)$ and then encodes with the nearest kernel, and the "compress" option `tc` uses global average pooling for all tail frames and compresses them with the nearest kernel.

*Skipping*: The notation `x` skips an arbitrary number of frames (including 0 frames).

*History Discretization*: The notation `+d` means converting history into discrete space.

*Generating*: The notation `g9` means generating 9 frames.

With the above naming convention, we can represent all ablative structures in a compact form. Note that this naming also implies the sampling approach as discussed in Section 4.1:

`td_f16k4f4k2f1k1_g9`: The vanilla sampling that generates frames in temporal order.

`td_f16k4f4k2f1k1_g9+D`: The vanilla sampling with history discretization.

`td_f16k4f4k2f1k1_g9_x_f1k1`: The anti-drifting sampling with an endpoint frame.

`f1k1_x_g9_f1k1f4k2f16k4_td`: The inverted anti-drifting sampling in inverted temporal order.

## 5.2 Base Model and Implementation Details

We implement FramePack with Wan and HunyuanVideo. We implement both the text-to-video and image-to-video structures, though both are naturally supported by next-frame-section prediction models and do not need architecture modifications. We report results with HunyuanVideo in the main paper (see also supplementary for Wan results). We conduct all experiments using H100 GPU clusters with training details in the supplementary. Note that FramePack achieves a batch size of about 64 on a single 8×A100-80G node with the 13B HunyuanVideo model at 480p resolution LoRA training with window size 2 or 3 (or batch size 32 of window size 4 or 5), making FramePack suitable for personal or laboratory-scale training and experimentation. We follow the guidelines of LTXVideo [17]'s dataset collection pipeline to gather data at multiple resolutions and quality levels (see also supplementary for more details).

## 5.3 Quantitative Evaluation

We discuss the metrics for evaluating ablative architectures. The tested inputs consist of 512 real user prompts for text-to-video and 512 image-prompt pairs for image-to-video tasks. All test samples were curated from real users to ensure diversity and real-world applicability. For quantitative tests, we by default use 30 seconds for long videos and 5 seconds for short videos.

**Metrics** Multiple metrics for video evaluations are consistent with common benchmarks, *e.g.*, VBench [24], VBench2 [75], *etc*. *Clarity*: The MUSIQ [27] image quality predictor trained on SPAQ [14]. This metric measures artifacts like noise and blurring. *Aesthetic*: The LAION aesthetic predictor [40]. This metric measures the aesthetic values perceived by a CLIP-based estimator. *Motion*: The video frame interpolation model [23] modified by VBench to measure the smoothness of motion. *Dynamic*: The RAFT [45] modified by VBench to estimate the degree of dynamics. Note that the "dynamic" metric and "motion" metric represent a trade-off, *e.g.*, a still image may rank high on motion smoothness but will be penalized by low dynamic degrees. *Semantic*: The video-text score computed by ViCLIP [53]. This metric measures the overall semantic consistency between the generated video and the prompt. *Anatomy*: The ViT [13] pretrained by VBench for identifying the per-frame presence of hands, faces, bodies, *etc*. *Identity*: The facial feature similarity using ArcFace [11] with face detection by RetinaFace [12].

**Drifting measurement** We observe that when drifting occurs, a significant difference emerges between the beginning and ending portions of a video across various quality metrics. We define the start-end contrast $\Delta_{\text{drift}}^M$ for an arbitrary quality metric $M$ as:

$$\Delta_{\text{drift}}^M(V) = |M(V_{\text{start}}) - M(V_{\text{end}})| \,, \tag{7}$$

where $V$ is the tested video, $V_{\text{start}}$ represents the first 15% of frames, and $V_{\text{end}}$ represents the last 15% of frames. This start-end contrast can be applied to different metrics $M$ (*e.g.*, motion score, image

Table 1: **Ablation study.** We evaluate different FramePack configurations across multiple global metrics, drifting metrics, and human assessments. The table is divided into 4 groups based on sampling approach: vanilla sampling, anti-drifting sampling, and inverted anti-drifting sampling, and vanilla sampling with discrete history. The tests are conducted with HunyuanVideo as base. Bests in bold. ELO differences within ±16 are considered ties.

| Variant | Global Metrics ↑ | | | | | | | Drifting Metrics ↓ | | | | Human | |
|---|---|---|---|---|---|---|---|---|---|---|---|---|---|
| | Clarity | Aesthetic | Motion | Dynamic | Semantic | Anatomy | Identity | $\Delta_{\text{drift}}^{\text{Clarity}}$ | $\Delta_{\text{drift}}^{\text{Motion}}$ | $\Delta_{\text{drift}}^{\text{Semantic}}$ | $\Delta_{\text{drift}}^{\text{Anatomy}}$ | ELO ↑ | Rank* ↓ |
| td_f8k8f4k4f2k2f1k1_g9 | 67.33% | 65.94% | 94.53% | 92.91% | 20.06% | 66.97% | 71.72% | 3.25% | 3.45% | 7.45% | 16.56% | 1090 | 5 |
| td_f64k8f16k4f4k2f1k1_g9 | 67.71% | 66.71% | 95.09% | 93.91% | 21.39% | 66.56% | 71.50% | 3.22% | 3.38% | 7.25% | 16.43% | 1070 | 5 |
| td_f512k8f64k4f8k2f1k1_g9 | 67.74% | 66.44% | 94.27% | 92.91% | 21.90% | 67.73% | 71.35% | 3.18% | 3.32% | 7.05% | 15.95% | 1085 | 5 |
| td_f512k16f64k8f8k2f1k1_g9 | 66.13% | 64.52% | 94.77% | 94.18% | 19.21% | 65.72% | 71.51% | 3.25% | 3.45% | 7.85% | 17.62% | 1068 | 5 |
| td_f16k4f4k2f1k1_g9 | 67.37% | 65.05% | 95.97% | 91.66% | 19.15% | 65.38% | 71.73% | 3.22% | 3.48% | 6.65% | 17.36% | 1072 | 5 |
| td_f16k4f2k2f1k1_g1 | 65.81% | 64.39% | 94.91% | **94.92%** | 19.20% | 64.16% | 69.70% | 3.43% | 3.77% | 9.81% | 20.89% | 1030 | 6 |
| td_f16k4f2k2f1k1_g4 | 66.57% | 64.99% | 94.00% | 82.85% | 19.40% | 65.97% | 69.06% | 3.36% | 3.68% | 8.55% | 19.09% | 1050 | 6 |
| td_f16k4f2k2f1k1_g9 | 67.15% | 65.29% | 94.08% | 92.97% | 20.73% | 66.46% | 71.08% | 3.18% | 3.42% | 7.45% | 18.05% | 1074 | 5 |
| tc_f16k4f2k2f1k1_g9 | 67.62% | 65.07% | 94.02% | 90.33% | 21.71% | 71.70% | 71.01% | 3.15% | 3.25% | 7.25% | 17.21% | 1088 | 5 |
| ta_f16k4f2k2f1k1_g9 | 67.02% | 66.44% | 94.11% | 91.09% | 21.48% | 71.92% | 72.18% | 3.12% | 3.18% | 7.05% | 17.66% | 1092 | 5 |
| td_f8k8f4k4f2k2f1k1_g9_x_f1k1 | 68.46% | 66.95% | 96.46% | 75.55% | 22.88% | 85.10% | 75.84% | 2.95% | 2.85% | 5.95% | 15.87% | **1135** | 3 |
| td_f64k8f16k4f4k2f1k1_g9_x_f1k1 | 68.21% | 67.75% | 95.74% | 85.97% | 22.79% | 81.09% | 76.29% | 2.92% | 2.98% | 5.95% | 15.99% | **1140** | 3 |
| td_f512k8f64k4f8k2f1k1_g9_x_f1k1 | 68.62% | 67.72% | 95.05% | 76.50% | 22.44% | 82.01% | 76.65% | 2.88% | 2.92% | 5.75% | 15.94% | 1118 | 4 |
| td_f512k16f64k8f8k2f1k1_g9_x_f1k1 | 68.35% | 67.89% | 97.73% | 76.48% | 22.75% | 78.40% | 76.16% | 2.85% | 2.85% | 5.55% | 14.04% | 1115 | 4 |
| td_f16k4f4k2f1k1_g9_x_f1k1 | 68.42% | 67.59% | 96.74% | 74.37% | 23.69% | 79.14% | 77.37% | 2.82% | 2.78% | 5.35% | 14.90% | 1138 | 3 |
| td_f16k4f2k2f1k1_g1_x_f1k1 | 65.32% | 67.97% | 95.26% | 81.69% | 19.97% | 74.57% | 77.93% | 2.78% | 2.72% | 4.95% | 14.80% | 1080 | 5 |
| td_f16k4f2k2f1k1_g4_x_f1k1 | 69.92% | 67.49% | 97.84% | 71.90% | 21.12% | 74.84% | 77.53% | 2.75% | 2.65% | 4.95% | 14.98% | 1100 | 4 |
| td_f16k4f2k2f1k1_g9_x_f1k1 | 69.51% | **69.15%** | 96.97% | 77.41% | 23.03% | 83.10% | 69.25% | 2.72% | 2.58% | 4.75% | 13.73% | 1142 | 3 |
| tc_f16k4f2k2f1k1_g9_x_f1k1 | 69.62% | 68.42% | 96.45% | 82.27% | 23.08% | 81.68% | 69.08% | 2.68% | 2.52% | 4.55% | 13.54% | 1145 | 3 |
| ta_f16k4f2k2f1k1_g9_x_f1k1 | 69.21% | 68.84% | 97.87% | 76.22% | 22.77% | 81.70% | 75.23% | 2.65% | 2.45% | 4.35% | 13.76% | **1150** | 3 |
| f1k1_x_g9_f1k1f2k2f4k4f8k8_td | 69.62% | 67.87% | 97.93% | 88.79% | 23.73% | 86.99% | 78.63% | 2.55% | 2.35% | 4.15% | 12.71% | **1210** | **1** |
| f1k1_x_g9_f1k1f4k2f16k4f64k8_td | 69.56% | 67.48% | 97.42% | 86.68% | 24.48% | 86.35% | 78.06% | 2.45% | 2.25% | 3.95% | 12.52% | **1215** | **1** |
| f1k1_x_g9_f1k1f8k2f64k4f512k8_td | 69.35% | 67.89% | 97.85% | 88.21% | 24.66% | 76.99% | 79.56% | 2.35% | 2.15% | 3.75% | 12.13% | **1220** | **1** |
| f1k1_x_g9_f1k1f8k2f64k8f512k16_td | 69.20% | 68.76% | **99.11%** | 89.18% | 24.35% | 77.62% | 79.88% | 2.35% | 2.05% | 3.55% | 11.10% | **1235** | **1** |
| f1k1_x_g9_f1k1f4k2f16k4_td | 69.25% | 67.40% | 98.59% | 89.01% | 24.24% | 77.31% | 79.16% | 2.45% | 1.95% | 3.35% | **9.22%** | **1225** | **1** |
| f1k1_x_g1_f1k1f2k2f16k4_td | 69.74% | 67.04% | 98.09% | 79.85% | 24.89% | 77.27% | 80.86% | 2.55% | 2.15% | 3.75% | 11.37% | 1150 | 3 |
| f1k1_x_g4_f1k1f2k2f16k4_td | 70.28% | 68.11% | 98.30% | 79.45% | 24.87% | 77.95% | 80.23% | 2.45% | 2.05% | 3.75% | 11.12% | 1175 | 2 |
| f1k1_x_g9_f1k1f2k2f16k4_td | 70.73% | 67.97% | 98.11% | 88.76% | 25.79% | 78.45% | **84.01%** | 2.35% | 1.95% | 3.65% | 11.98% | **1228** | **1** |
| f1k1_x_g9_f1k1f2k2f16k4_tc | 70.48% | 68.74% | 98.16% | 89.18% | **27.01%** | 87.21% | 81.04% | **2.18%** | 1.85% | **2.54%** | 11.71% | **1230** | **1** |
| f1k1_x_g9_f1k1f2k2f16k4_ta | **70.81%** | 67.31% | 98.15% | 80.72% | 25.60% | 85.60% | 82.76% | 2.25% | **1.77%** | 2.95% | 9.85% | **1232** | **1** |
| td_f8k8f4k4f2k2f1k1_g9+D | 69.12% | 66.53% | 96.91% | 93.07% | 24.79% | 82.58% | 73.34% | 2.43% | 2.74% | 4.45% | 14.37% | **1225** | **1** |
| td_f64k8f16k4f4k2f1k1_g9+D | 69.35% | 65.99% | 97.21% | 92.83% | 22.43% | 81.88% | 72.90% | 2.22% | 2.44% | 4.49% | 13.74% | 1170 | 2 |
| td_f512k8f64k4f8k2f1k1_g9+D | 69.26% | 67.24% | 96.28% | 91.01% | 24.63% | 82.58% | 71.07% | 2.26% | 2.01% | 5.46% | 13.00% | 1169 | 2 |
| td_f512k16f64k8f8k2f1k1_g9+D | 69.93% | 67.26% | 97.60% | 93.61% | 23.87% | 81.27% | 73.39% | 2.93% | 2.92% | 5.11% | 15.70% | 1139 | 3 |
| td_f16k4f4k2f1k1_g9+D | 69.49% | 67.25% | 96.95% | 92.39% | 23.40% | **88.67%** | 72.08% | 2.30% | 2.63% | 6.74% | 13.51% | **1223** | **1** |
| td_f16k4f2k2f1k1_g1+D | 68.16% | 65.47% | 95.94% | 93.15% | 23.97% | 75.24% | 71.78% | 2.66% | 3.70% | 6.40% | 16.83% | 1145 | 3 |
| td_f16k4f2k2f1k1_g4+D | 68.16% | 65.80% | 95.26% | 92.97% | 22.97% | 69.87% | 71.91% | 2.70% | 3.73% | 5.69% | 17.06% | 1142 | 3 |
| td_f16k4f2k2f1k1_g9+D | 69.78% | 67.22% | 96.91% | 91.39% | 24.40% | 77.62% | 74.95% | 2.30% | 2.49% | 4.12% | 14.11% | **1222** | **1** |
| tc_f16k4f2k2f1k1_g9+D | 69.42% | 66.24% | 96.01% | 92.53% | 23.06% | 79.15% | 72.76% | 2.87% | 2.43% | 5.51% | 13.67% | **1218** | **1** |
| ta_f16k4f2k2f1k1_g9+D | 68.51% | 67.77% | 96.90% | 93.92% | 23.24% | 79.90% | 70.01% | 2.27% | 2.41% | 4.31% | 15.85% | **1216** | **1** |

* Based on the ELO scores and tie rules, the ranks are divided into: 1030-1050 (Rank 6), 1068-1092 (Rank 5), 1100-1118 (Rank 4), 1135-1150 (Rank 3), 1169-1175 (Rank 2), and 1210-1235 (Rank 1).

quality, *etc*.). The magnitude of $\Delta_{\text{drift}}^M(V)$ directly indicates the severity of drifting. Since video models may generate frames in different temporal orders (either forward or backward), we use the absolute difference to ensure our metric remains direction-agnostic.

**Human assessments**  We collect human preferences from A/B tests. Each ablative architecture yields 100 results. The A/B tests are randomly distributed among ablations, and we ensure that each ablation covers at least 100 assessments. We report ELO-K32 score and the relative ranking.

**Ablative results**  As shown in Table 1, we note several discoveries. (1) The inverted anti-drifting sampling method achieves the best results in 4 out of 7 metrics, and achieves the best performance in all drifting metrics. (2) However, the inverted anti-drifting sampling has a relatively small dynamic range. (3) While vanilla sampling achieved the highest dynamic score, this is likely attributable to drifting effects rather than genuine quality, evidenced by relatively low ELO scores. (4) The vanilla sampling with discrete history achieves highly competitive human scores while having a much larger dynamic range. (5) We also observe that differences between specific configuration options within the same sampling approach are relatively small and random, suggesting that the overall architecture contributes more to the general difference.

**History discretization parameter**  The history discretization is influenced by the parameter $K$ with higher $K$ giving stronger anti-drifting effects but also more challenging learning for smooth transitions between sections. In our tests, $K = 128$ gives strong drift reduction with relatively minimal training difficulties. We provide more detailed ablations for $K$ in the supplementary materials.

**Additional evaluations**  We test feature-similarity-based packing with video game (world model) benchmarks in the supplementary material. See also supplementary material for Wan results and more details of decomposed metrics.

Table 2: **Comparison with relevant methods.** We compare across the same global metrics, drifting metrics, and human assessments. The tests are conducted with HunyuanVideo as base. Bests in bold. ELO differences within $\pm 16$ are considered ties.

| Method | Global Metrics ↑ | | | | | | | Drifting Metrics ↓ | | | | Human | |
|---|---|---|---|---|---|---|---|---|---|---|---|---|---|
| | Clarity | Aesthetic | Motion | Dynamic | Semantic | Anatomy | Identity | $\Delta_{\text{drift}}^{\text{Clarity}}$ | $\Delta_{\text{drift}}^{\text{Motion}}$ | $\Delta_{\text{drift}}^{\text{Semantic}}$ | $\Delta_{\text{drift}}^{\text{Anatomy}}$ | ELO ↑ | Rank ↓ |
| Repeating image-to-video | 56.73% | 56.15% | 94.34% | 91.21% | 17.74% | 69.41% | 73.06% | 9.51% | 3.92% | 9.95% | 19.88% | 1015 | 5 |
| Anchor frames (resembling StreamingT2V [20]) | 69.58% | 67.35% | **99.96%** | 74.97% | 25.76% | 85.01% | 79.52% | 2.85% | 2.15% | 3.45% | 9.25% | 1173 | 2 |
| Causal attention (resembling CausVid [65]) | 62.88% | 59.41% | 96.98% | 88.27% | 19.15% | 72.74% | 75.32% | 7.45% | 3.15% | 6.75% | 15.96% | 1087 | 4 |
| DiffusionForcing [6] ($\sigma_{\text{train}}$ random, $\sigma_{\text{test}} = 0.1$) | 66.08% | 65.76% | 96.32% | 91.59% | 23.14% | 75.93% | 74.47% | 4.84% | 2.54% | 3.33% | 10.99% | 1170 | 2 |
| DiffusionForcing [6] ($\sigma_{\text{train}}$ random, $\sigma_{\text{test}} = 0.5$) | 67.41% | 66.66% | 92.93% | 91.08% | 24.03% | 77.83% | 76.42% | 3.55% | 2.39% | 3.48% | 9.40% | 1174 | 2 |
| DiffusionForcing [6] ($\sigma_{\text{train}}$ random, $\sigma_{\text{test}} = 0$) | 66.99% | 64.73% | 96.47% | 90.45% | 21.09% | 80.62% | 78.77% | 8.41% | 3.80% | 8.47% | 17.44% | 1095 | 4 |
| DiffusionForcing [6] ($\sigma_{\text{train}} = 0.1$, $\sigma_{\text{test}} = 0.1$) | 66.19% | 68.60% | 94.89% | **91.89%** | 22.49% | 76.12% | 78.27% | 6.82% | 3.79% | 5.08% | 10.78% | 1149 | 3 |
| History guidance (resembling HistoryGuidance [41]) | 68.05% | 68.74% | 97.01% | 73.39% | 24.88% | 81.84% | **83.42%** | 7.35% | 2.21% | 5.25% | 12.78% | 1152 | 3 |
| Inverted anti-drifting (f1k1_x_g9_f1k1f2k2f16k4) | **71.15%** | 68.71% | 99.45% | 89.29% | 28.15% | **86.53%** | 82.11% | **2.25%** | **1.85%** | **2.68%** | **8.58%** | 1220 | **1** |
| Vanilla + discrete history (f16k4f2k2f1k1_g9+D, $K = 256$) | 70.01% | **68.76%** | 95.65% | 91.74% | **28.37%** | 86.41% | 82.22% | 3.13% | 2.05% | 2.89% | 8.74% | **1224** | **1** |

## 5.4 Comparison to Alternative Architectures

We discuss several relevant alternatives to generate videos in various ways. The involved methods either enable longer video generation, reduce computational bottlenecks, or both. To be specific, we implement these variants on top of HunyuanVideo default architecture (33 latent frames) using a simple naive sliding window with half context length for history inputs.

*Repeating image-to-video*: Directly repeat the image-to-video inference to make longer videos.

*Anchor frames*: Use an image as the anchor frame to avoid drifting. We implement a structure that resembles StreamingT2V [20].

*Causal attention*: Finetune full attention into causal attention for easier KV cache and faster inference. We implement a structure that resembles CausVid [65].

*DiffusionForcing*: We conduct a detailed ablation study with DiffusionForcing [6]. We focus on the history noise scheduling using the same scheduling as SkyreelV2 [7] that multiplies the diffusion timestep on history frames with $\sigma_{\text{train}}$ during training and $\sigma_{\text{test}}$ in inference to delay the denoising on history latents. Intuitively, $\sigma_{\text{test}} = 0$ is equivalent to clean latents for history (no noise added to the history). We consider these ablations: (1) $\sigma_{\text{train}}$ being random with $\sigma_{\text{test}} = 0.1$; (2) $\sigma_{\text{train}}$ being random with $\sigma_{\text{test}} = 0.5$; (3) $\sigma_{\text{train}}$ being random with $\sigma_{\text{test}} = 0$; (4) $\sigma_{\text{train}} = 0.1$ and $\sigma_{\text{test}} = 0.1$. Usually higher $\sigma_{\text{test}}$ reduces the reliance on the history, which is beneficial for interrupting error accumulation, thus mitigates drifting, but at the cost of aggravating forgetting.

*History guidance*: Delay the denoising timestep on history latents but also put the completely noised history on the unconditional side of CFG guidance. This will speed up error accumulation, thus aggravating drifting, but also enhance memory to mitigate forgetting. We implement a structure that resembles HistoryGuidance [41].

As shown in Table 2, we observe several findings. (1) The inverted anti-drifting sampling achieves the best results across all drifting metrics, while having a relatively small dynamic range. (2) The vanilla sampling with discrete history is very competitive in drifting measurements, while having a relatively larger dynamic range. (3) Human perception prefers the two proposed candidates as evidenced by the ELO score. (4) See also the supplementary material for more detailed explanations, analysis, and comparisons with DiffusionForcing candidates.

## 6 Conclusion

In this paper, we presented FramePack, a neural network structure that aims to address the forgetting-drifting dilemma in next-frame prediction models for video generation. FramePack applies progressive compression to input frames based on their importance, ensuring the context length converges to a fixed upper bound. We discussed both time-proximity-based packing and feature-similarity-based packing. We also discussed the anti-drifting training methods with history discretization, and the anti-drifting sampling methods using bi-directional context planning and scheduling. Experiments suggest that FramePack can process a large number of frames, improve model responsiveness, and allow for higher batch sizes in training. The approach is compatible with existing video diffusion models and supports various compression variants that can be optimized for wider applications.

## Acknowledgments and Disclosure of Funding

This work was partially supported by the Brown Institute for Media Innovation, by Google through their affiliation with Stanford Institute for Human-centered Artificial Intelligence (HAI) and by a Hoffman-Yee HAI grant.

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
