# OpenReview forum: "Frame Context Packing and Drift Prevention in Next-Frame-Prediction Video Diffusion Models"
_NeurIPS.cc/2025/Conference — NeurIPS 2025 spotlight_

### Official Review · Reviewer_ZncW · 2025-06-30

**Clarity:** 4
**Significance:** 4
**Originality:** 4
**Rating:** 6
**Confidence:** 5

**Summary:**

This paper investigates the next-frame prediction problem and introduces efficient frame context packing techniques that enable conditioning on thousands of frames during inference and support training with very large batch sizes. To tackle the issue of error accumulation over long sequences, it also proposes several drift prevention strategies. Specifically, the authors explore various packing strategies, including time proximity-based and feature similarity-based methods, and examine numerous variants in packing schedules, patchification levels, tail handling options, and positional embedding alignment. For drift prevention, they study endpoint planning, alternative sampling orders, and apply history discretization to reduce the distribution gap between training and inference. Overall, this work presents a comprehensive study of frame context compression and anti-drifting techniques, achieving strong results demonstrated by compelling video examples. The proposed methods are thoroughly examined and validated through extensive ablation experiments and comparisons against existing approaches across multiple benchmarks.

**Questions:**

I don't have any other questions.

**Ethical Concerns:**

["NO or VERY MINOR ethics concerns only"]

**Final Justification:**

Based on all the reviews and the authors' response, I have decided to keep my rating of Strong Accept for this paper.

**Limitations:**

Yes.

**Paper Formatting Concerns:**

N/A.

**Quality:**

4

**Strengths And Weaknesses:**

Strengths:
1. The paper addresses a highly practical and important challenge, enabling video diffusion generation with very large temporal context while preventing drifting. The proposed FramePack method, along with its extensive exploration of design choices and variants, is both novel and insightful.
2. I especially appreciate the discussion on the fundamental dilemma between forgetting and drifting. The paper clearly articulates how existing approaches often trade one for the other, and how the observation of substantial duplication of visual features across nearby frames naturally motivates the efficient packing strategy.
3. The experimental study is exceptionally thorough. It not only covers the key innovations, such as time proximity-based, feature similarity-based, and hybrid packing strategies, endpoint planning, sampling order adjustments, and history discretization, but also rigorously investigates many design variants, including packing schedules, patchification levels, and anti-drifting sampling orders. The method is tested on two popular video diffusion backbones (Wan and Hunyuan), across diverse datasets (VBench, TECO DMLab, MineCraft, UCF-101), with quantitative metrics and human assessments.
4. The paper compares carefully against multiple relevant methods from recent literature, including anchor-frame approaches (StreamingT2V), causal attention (CausVid), Diffusion Forcing, and HistoryGuidance. These comparisons convincingly demonstrate the advantages of the proposed techniques.
5. The proposed framework is highly computationally efficient, making it feasible for personal or academic-scale experimentation. The clear implementation details and open-sourced code will likely have a significant impact on the community, lowering barriers for future work in this important area.
6. The video demos provided in the supplemental materials are of exceptional quality, strongly validating the effectiveness of the approach and highlighting its potential for real-world applications.
7. Despite introducing many intertwined ideas and technical components, the paper is very well organized. All methods are clearly described, justified, and discussed, making this a very solid and convincing contribution.

----------------------------------

I did not find any critical weaknesses in this paper. The only minor concern is that the proposed packing and anti-drifting strategies are largely hand-crafted and manually selected. However, this limitation is acknowledged by the authors in Section K, and the effectiveness of their choices is well demonstrated by the video results.

---

> ### Author Rebuttal · Authors · 2025-07-29
>
> Thanks for the evaluations and insights. The tests related to packing and anti-drifting strategies are designed to exploit potential solutions and their internal connections or influences. We have plans to expand the tests to include even more strategies or models (like models that are released after the submission.)

---

### Official Review · Reviewer_LWB1 · 2025-07-02

**Clarity:** 3
**Significance:** 3
**Originality:** 4
**Rating:** 4
**Confidence:** 5

**Summary:**

This paper introduces FramePack, a novel method that removes temporal redundancy in video generation by constructing a binary pyramid structure. This compression addresses the challenge of balancing memory retention and mitigating visual degradation over time.

**Questions:**

- How is the estimated next frame section $\hat x$ obtained to compute the similarity score during inference, particularly at the first denoising steps when no estimation exists yet? Is the history frame order updated during multiple denoising steps as the estimation improves?
- The Drift Prevention component proposes various sampling strategies for different scenarios, but these appear similar to hierarchical video generation methods such as NUWA-XL [3] and Hierarchy-2 of [2]. Could the authors clarify the advantages of their method compared to these approaches?
- Can the authors provide results that demonstrate the method's applicability to videos with more dynamic scenes, beyond the periodic or repetitive cases shown in the supplementary materials?
- Could the authors visualize the activated attention maps to verify that the model does not overly rely on the last few frames and effectively attends to the entire history sequence?

[3] Yin et al., NUWA-XL: Diffusion over Diffusion for eXtremely Long Video Generation, ACL 2023.
[4] Harvey et al., Flexible Diffusion Modeling of Long Videos, NeurIPS 2022.

**Ethical Concerns:**

["NO or VERY MINOR ethics concerns only"]

**Final Justification:**

I've decided to keep my score after rebuttal process.

**Limitations:**

Yes. The authors addressed the limitations of their work.

Common comment for the authors: If I couldn't fully understand your methods or am underestimating your contributions, feel free to explain and let's discuss. I will gladly raise the scores if I have misunderstood key points or if my concerns are addressed well.

**Quality:**

4

**Strengths And Weaknesses:**

### Strengths
- Presents a novel method for effectively reducing temporal redundancy in videos through a binary pyramid approach.
- The paper is well-written, with a clear problem definition, informative figures, and sound formulations.
- Provides solid and comprehensive experimental analysis to support the proposed method.

### Weakness
- The practical usage appears limited. The supplementary videos are mainly repetitive or periodic scenes with near-static background. It remains unclear whether the method can handle more dynamic scenes as suggested in L166-170.
- The packing strategy merely based on temporal order and similarity might introduce redundancy into $F$. There is concern that the model may learn to copy $F_{i\leq k}$ by attending only to them, rather than making use of $F_{i>k}$ for small $k$ (e.g. 2~5). Visualization of the activated attention would help confirm that all historical frames are effectively utilized. Furthermore, I think this kind of strategy cannot extract important information from previous frames which are visually different with the current frame.
- The method relies on 3D attention models, which makes it challenging to integrate with approaches that use decoupled spatial and temporal attention. Most of the initial approaches for video diffusion models [1] and a SOTA approach recently exhibited [2] tried to train the video models efficiently using decoupled attention. FramePack's artificial packing may misalign temporal attention in such cases. However, this is likely a minor issue since many models still employ 3D attention.

[1] Blattmann et al., Align your Latents: High-Resolution Video Synthesis with Latent Diffusion Models, CVPR 2023.
[2] ByteDance Seed, Seedance 1.0: Exploring the Boundaries of Video Generation Models, arXiv 2025

---

> ### Author Rebuttal · Authors · 2025-07-30
>
> Thanks for the insightful and constructive comments.
>
> W1. The endpoint planning would give more results that are not repetitive or periodic, and some pure text-to-video cases would have even more dynamic. We are using 70-second stress tests for both cases and will add to this submission when revised.
>
> W2. Thanks for the suggestion for visualizing attention activations and we will implement that! Intuitively, one may think about some strong cases where the model must use all input frames to determine the motion – for example a car moving fast with many local high-frequency bumps while the distant skyscrapers are moving slowly with global low-frequency patterns, then we will need more frames to capture both high-frequency and low-frequency temporal details. In many cases, the model will need to utilize as many frames as possible.
>
> W3. Context compression is independent from the internal structure of models. If using some earlier models like UNet with decoupled spatial-temporal attentions, the context compression is equivalent to connecting some contexts directly to higher UNet levels. In more recent works like Seedance, the combability of spatial-temporal attention module is resolved internally (so it looks/works just like a DiT with 3D attention from outside), and this would not cause combability issues.
>
> Q1. Yes, the estimation is improved in the denoising steps. The first step uses the default score over all frames (but only the similarity part uses the default score – we still have the valid time score in the first step). But this design may cause some design choices related to teacache implementation.
>
> Q2. Thanks for the refences and we will add them! The NUWA-XL should be a model that generate long videos directly (not autoregressive or frame prediction). The Harvey et al. does have autoregressive options and frame prediction/planning schedules, and we will try the “Hierarchy-2” and add some numerical results. Intuitively, the Hierarchy-2 models longer range with dilatation and/or propagation (while framepack models this with compression and propagation). Thanks again for this comment.
>
> Q3. Yes. We implemented the results (W1) and will add to the submission when revised.
>
> Q4. Yes. We will implement the experiment mentioned in W2.

---

### Official Review · Reviewer_JF1b · 2025-07-02

**Clarity:** 3
**Significance:** 4
**Originality:** 4
**Rating:** 5
**Confidence:** 4

**Summary:**

This paper proposes FramePack, a next-frame prediction method that progressively compresses frame contexts based on their importance. It also introduces a series of novel anti-drifting methods using history discretization to prevent error accumulation. Extensive quantitative and qualitative experiments on the Hunyuan and Wan show that FramePack can process thousands of frames without any quality degradation. Moreover, it supports larger batch sizes and enables efficient long-context next-frame post-training.

**Questions:**

See weakness

**Ethical Concerns:**

["NO or VERY MINOR ethics concerns only"]

**Final Justification:**

My concerns are well resolved, so I keep my original score.

**Limitations:**

yes

**Paper Formatting Concerns:**

no formatting concerns

**Quality:**

4

**Strengths And Weaknesses:**

**Strengths**
1. The FramePack architecture serves as a robust memory by combining both temporal-proximity and feature-level permutations for packing, ensuring reliable context selection. Furthermore, its time-proximity based packing is shown to be independent of context length, allowing fixed-length compression and removing the need for an attention KV cache—resulting in a highly efficient design.

2. Novel inverted anti-drifting: By first predicting the first and last frames and then reconstructing intermediate frames in reverse temporal order, the method effectively mitigates drift and error accumulation. As shown in Table 1, this inverted prediction scheme yields the best performance in reducing long-term prediction errors.

3. History discretization: Replacing continuous frame histories with discrete integer tokens offers a compact, loss-aware representation of past contexts. Supplementary experiments demonstrate that this discretization retains essential information and further prevents error accumulation.

**Weaknesses**

1. The ablation names in Table 1 are overly complex; using simpler labels such as "vanilla sampling" and "inverted anti-drifting sampling" would make the study easier for readers to follow.

2. Drift is measured only on the first and last 15% of frames, which—while efficient—may miss errors accumulating in the middle of sequences; quantitative drift measurement itself remains a challenging open problem.

3. It is unclear whether history discretization could be combined with the inverted anti-drifting scheme for additional gains, as this potential synergy is not investigated.

4. The abstract claims that the anti-drifting methods also work in single-direction video streaming, but no concrete experiments or results are provided to support this.

5. The current FramePack relies on manually designed feature selection; a dynamic, content-adaptive context compression strategy could enable multi-shot videos with richer narrative variation rather than repetitive motion.

6. Overall, the paper more like an extensive empirical report and would benefit from more direct, high-level conclusions, i.e., clear conclusion which support by the rich ablation results.

---

> ### Author Rebuttal · Authors · 2025-07-30
>
> Thanks for the constructive evaluations.
>
> 1. We will revise the labels as suggested.
>
> 2. Thanks for this comment! Quantitative drift measurement is indeed difficult, and we will add some manuscript clarifications, and maybe also show some numeric results of the mentioned middle part.
>
> 3. History discretization can be combined with inverted anti-drifting. But the inverted sampling by itself is very strong regulation and the influence of history discretization should be less obvious in this combination.
>
> 4. Thanks for mentioning this, we will add more single-direction video streaming results. This would be the combination with only history discretization, or with planning but planned on-the-fly.
>
> 5. Thanks for the suggestion. We are investigating the possibilities to learn the packing or use some more dynamic routing approaches.
>
> 6. Thanks for the insight – there are some (probably less salient) high-level take-aways like “packing and planning deserve more research” or “forgetting v.s. drifting”, while extensive study and report did become more direct outcomes within these considerations.

---

### Official Review · Reviewer_47tZ · 2025-07-02

**Clarity:** 3
**Significance:** 3
**Originality:** 3
**Rating:** 5
**Confidence:** 5

**Summary:**

This paper proposes FramePack to address the dilemma of forgetting and drifting in next-frame video diffusion models. It improves memory efficiency by compressing past frames based on their importance, and mitigates drifting through techniques such as inverted sampling and history discretization. FramePack enables long video generation with low GPU memory usage, while maintaining high visual quality and temporal consistency.

**Questions:**

1.	The proposed inverted anti-drifting sampling generates frames in reverse temporal order to reduce error accumulation. However, does this approach compromise temporal realism or physical consistency, especially for motions governed by causality and physical laws? Are there specific motion types or scenarios where inverted sampling may be unsuitable?
2.	Many sampling strategies (e.g., endpoint planning, inverted sampling) rely on local patterns and may suit periodic or short clips. Can these methods handle long-range dependencies like narrative twists or foreshadowing?

**Ethical Concerns:**

["NO or VERY MINOR ethics concerns only"]

**Final Justification:**

My initial concerns are addressed, and I will keep a positive rating.

**Limitations:**

Yes. The authors provide a thorough and balanced evaluation in the experimental section, clearly addressing the strengths, limitations, and trade-offs of the different approaches.

**Quality:**

3

**Strengths And Weaknesses:**

Strengths
1.	The idea of compressing frame context based on frame importance (time proximity and feature similarity) is intuitive and elegant, enabling practical scaling to thousands of frames within a fixed context size.
2.	The method achieves high efficiency and can run on laptops with limited GPU memory (e.g., 6–8GB), making it appealing for both academia and industry.
3.	The method systematically explores a wide range of compression and anti-drifting sampling strategies, with thorough empirical validation that offers valuable insights into their strengths.
Weaknesses
1.	The use of heuristic-based importance measures (e.g., time proximity or latent similarity) may not generalize well to dynamic scenes, such as fast human actions, rapid camera movements, or sudden scene transitions. Aggressive compression or discretization could discard fine-grained temporal cues, which is not adequately analyzed in the paper.
2.	While strategies like endpoint planning or inverted sampling handle repetitive motions well, it remains unclear whether they can support storylines with foreshadowing, twists, or scene-level progression beyond local coherence.

---

> ### Author Rebuttal · Authors · 2025-07-30
>
> Thanks for the insightful and constructive comments.
>
> W1. Heuristic Measures and More Scenes. Thanks for the insight to study how compression would influence “temporal cues”. Intuitively, those large global motions like rapid movements should still have obviously salient tendency even after high compression, while smaller motion behaviors can be further investigated. We will add some results to display the mentioned cases.
>
> W2. Sampling Strategies and More Motions. The endpoint planning naturally supports storylines like scene-level progression (mostly by prompt travelling). A well-trained model should also be able to plan the endpoints even without decomposing prompts (but that will be more difficult). But the foreshadowing can be a very interesting consideration that may need a combination of endpoints and/or inverted sampling. We will add some storylines results and try to find some foreshadowing examples. Thanks again for this suggestion.
>
> Q1. In theory, in a perfectly aligned ideal case when the video length and the motion length and all conditions are all perfectly aligned, the inverted sampling should not cause compromised behaviors since it is a self-supported generative model. However, in practice, people usually play with unaligned conditions or video lengths (like “a man stand up from sofa” and ask for a 60 second video, or “a ball drops from table to floor” and ask for 60 seconds – maybe that drop only need 3 seconds), and these are the primary factors of physical/temporal realism considerations. Yet, these do bring challenges to framework designs.
>
> Q2. Overall, single inverted sampling works better for short or simple (maybe periodic) clips, while endpoint planning is more suitable for narrative twists. The parameters of endpoints (like their distances) can also be adjusted to facilitate complicated narrative twists. We empirically consider the endpoint planning as a way to improve long-range dependencies - another way is to add some extra condition frames (while that needs more selective metrics).

---

### Decision · Program_Chairs · 2025-09-17

**Decision:**

Accept (spotlight)

**Comment:**

This paper received unanimously positive reviews (1 weak accept, 2 accepts, and 1 strong accept).

Here is a brief summary of the positive reviews.
- Reviewers agreed that the paper addresses a highly practical and important challenge in long-video generation, and that the proposed methods are both novel and insightful (ZncW, LWB1, JF1b). The core idea of compressing frame context based on importance was described as "intuitive and elegant" (47tZ).

- The paper was praised for its exceptionally thorough experimental study. Reviewers highlighted the systematic exploration of a wide range of strategies (47tZ), the "solid and comprehensive experimental analysis" (LWB1), and the rigorous investigation of many design variants (ZncW). The careful comparisons against multiple relevant methods were also seen as a strength (ZncW).

- The proposed framework was noted for being highly computationally efficient. Reviewers pointed out that it can run on hardware with limited GPU memory, making it accessible (47tZ), and that its design removes the need for an attention KV cache, resulting in high efficiency (JF1b, ZncW).

- The introduction of novel anti-drifting methods was identified as a key strength. Specifically, the "inverted anti-drifting" sampling scheme was highlighted for effectively mitigating drift and error accumulation (JF1b). The "history discretization" technique was also noted as a strength for offering a compact representation of past contexts (JF1b).

The AC thus recommends to accept this paper as a Highlight.